# Advances in Halloysite Nanotubes–Polysaccharide Nanocomposite Preparation and Applications

**DOI:** 10.3390/polym11060987

**Published:** 2019-06-04

**Authors:** Yang Wu, Yongzhi Zhang, Junping Ju, Hao Yan, Xiaoyu Huang, Yeqiang Tan

**Affiliations:** 1State Key Laboratory of Bio-fibers and Eco-textiles, Collaborative Innovation Center for Marine Biomass Fibers, Materials and Textiles of Shandong Province, School of Materials Science and Engineering, Qingdao University, Qingdao 266071, China; wy921170920@163.com (Y.W.); zyz18919@163.com (Y.Z.); yanhao1287476167@163.com (H.Y.); 2Key Laboratory of Synthetic and Self-Assembly Chemistry for organic Functional Molecules, Shanghai Institute of Organic Chemistry, Chinese Academy of Sciences, Shanghai 200032, China; xyhuang@sioc.ac.cn

**Keywords:** halloysite nanotubes, polysaccharide, interfacial interactions, reinforcing, adsorption

## Abstract

Halloysite nanotubes (HNTs), novel 1D natural materials with a unique tubular nanostructure, large aspect ratio, biocompatibility, and high mechanical strength, are promising nanofillers to improve the properties of polymers. In this review, we summarize the recent progress toward the development of polysaccharide-HNTs composites, paying attention to the main existence forms and wastewater treatment application particularly. The purification of HNTs and fabrication of the composites are discussed first. Polysaccharides, such as alginate, chitosan, starch, and cellulose, reinforced with HNTs show improved mechanical, thermal, and swelling properties. Finally, we summarize the unique characteristics of polysaccharide-HNTs composites and review the recent development of the practical applications.

## 1. Introduction

Nanofillers recently have drawn extensive attention from academic and industrial fields due to their unique performance [1]. The traditional materials, such as black carbon, graphite, silica, and silicate, can significantly improve the mechanical properties, thermal stability, and permeability of various polymers [2,3]. Nowadays, clay mineral nanofillers with large aspect ratios, high strength, and relatively low density have attracted intense research interest [4]. Clay minerals, natural materials with proven biocompatibility and abundant storage, exhibit unique properties for various applications [5]. The majority of the research concerning clay minerals is devoted to kaolinite [6], montmorillonite [7], and illite [8]. In recent years, halloysite nanotubes (HNTs), 1D natural materials with a unique tubular nanostructure, large aspect ratio, biocompatibility, and high mechanical strength, have arisen as promising nanofillers to improve the properties of polymers [9,10].

Halloysite was first proposed by Berthier (1826) [11]. Raw halloysite, which is usually white, is exploited from natural sediments and is easily processed into powder. The sizes of halloysite depend on its specific geological deposit, as reported in the literature on the basis of microscopy [12] and scattering techniques [13]. It possesses several typical morphologies, such as spherical, sheet-like, and tubular particles due to the diversity of crystallization conditions and geological occurrence. Among them, the tubular structure is the most common and valuable [14]. The tubular structure is caused by lattice mismatch between adjacent silicone dioxide and aluminum oxide layers [15]. The molecular formula of HNTs is Al_2_Si_2_O_5_(OH)_4_·nH_2_O, where n represents hydration or dehydration. HNTs are hydrated when n equals 2 and are dehydrated when n equals 0 [16,17,18,19]. Compared with traditional nanofillers, such as carbon nanotubes (CNTs) [20] and boron nitride nanotubes (BNNTs) [21], HNTs have a prominent advantage, which is that they are far less expensive [22]. The length of HNTs ranges from 100 to 2000 nm, with the inner diameter from 10 to 30 nm and the outer diameter from 30 to 50 nm. In terms of functional groups, HNTs contain a large amount of hydroxyl groups situated between layers and on the surface, respectively. Due to the multi-layer structure, most of the hydroxyl groups are inner groups. In addition, the inner surfaces of the HNTs are positively charged, while the outer surfaces are negatively charged [12,23]. The detailed data of HNTs are listed in Table 1 [14,24].

Although the characteristics above generate excellent mechanical, thermal, and regenerable properties, the direct application of HNTs is limited. The drawbacks include difficulty in dissolving, brittleness, and low permeability [25]. With abundantly renewable sources and charming properties, including inherent biocompatibility, polysaccharides have attracted rising attention, and they have been widely applied to the medical [26], textile [27], and food fields, among others [28,29]. By preparing polysaccharide-HNTs composites, we can overcome these shortcomings. Due to the stable tubular morphology, charge distribution, the specific origin, and unique crystal structure, HNTs can be dispersed into single particles easily and the lumen diameter of HNTs fits well to macromolecule and protein diameters, causing the good combination between polysaccharides and HNTs [30,31,32]. The present research mainly focuses on alginate [33,34], chitosan [35], starch [36], cellulose [37], pectin, and carrageenan [38].

Although general properties of polysaccharide/halloysite nanotube composites and biomedical applications have been reviewed earlier by Liu et al. [39], we review the recent progress toward the development of polysaccharide-HNTs composites, paying attention to the main existence forms, wastewater treatment, and food packaging applications particularly. Through this review, we have a better understanding of unique characteristics of polysaccharide-HNTs composites, which can be helpful to the continuous expansion of their application in the future.

## 2. Preparation of Polysaccharide-HNTs Composites

### 2.1. Purification

Raw halloysite has impurities, such as quartz, illite, and perlite, since it is exploited directly from natural deposits. Therefore, the aggregate nanotubes should be separated to purify the HNTs before use in practical applications [40]. The traditional method of purification is the dispersion-centrifugation-drying technique. Firstly, we slowly added HNTs powder into deionized water under heating and mild stirring conditions. Then, the solution was further processed by lavation with deionized water three times and centrifugation. Finally, the pure HNTs were obtained after desiccation [41]. Figure 1 showed FE-SEM (Left) photos of HNTs and schematic illustration of crystalline structure (Right) of HNTs.

### 2.2. HNTs/Polysaccharide Preparations and Formulations

Using traditional processing techniques, HNTs can be mixed with most polysaccharides, such as alginate, chitosan, starch, cellulose, and carrageenan. The purpose of different fabrication methods is to enhance the interfacial interactions and dispersibility. In this section, we introduce the main existence forms of polysaccharide-HNTs composites.

#### 2.2.1. Hydrogels

The hydrophilic structure of hydrogels enables them to hold large amounts of water in the three-dimensional networks. Due to the characteristics of high hygroscopicity and low stiffness, hydrogels are usually described as soft and wet materials [43,44]. Chan et al. prepared a HNTs/alginate hydrogel and the effects of HNTs on the physicochemical, thermal, mechanical, and mass transfer properties of alginate hydrogel beads were investigated in detail [45]. It was found that HNTs filled the interspace in the alginate matrix and allowed more efficient load transfer. The HNTs were embedded in the layers of alginate hydrogel networks and they had little effect on the size and on the shape of the alginate beads. The mechanism for enhanced mechanical strength could be attributed to physical interaction between the alginate and HNTs, and the mechanical strength could be improved at lower HNTs loading if chemical interactions were present. Zhou et al. reported alginate/HNTs composite hydrogels via solution mixing and subsequent cross-linking with calcium ions [46]. The static and shear viscosity of composite solutions increases with the increase of HNTs. The rheological behaviors of alginate/HNTs solutions were a shear thinning and fit with the power law model. Due to the good dispersion ability of HNTs, polysaccharides and HNTs are mixed easily via interfacial interactions, such as electrostatic and hydrogen bonding interactions, contributing to the formation of homogeneous composites and enhanced properties. Fourier-transform infrared spectroscopy (FTIR) and X-ray powder diffraction (XRD) are applied to study the interfacial interactions between alginate and HNTs. As shown in Figure 2b, the peaks at 1419 cm^−1^ shifted to higher wave numbers and no new peaks appeared in the composites, which indicated that hydrogen bond interactions occur between HNTs and alginate but no chemical reaction occurs. The XRD patterns of composites (Figure 2c) were very similar to HNTs no new diffraction peak occurring, which suggested the crystal structure of HNTs was retained in the composites.

The effect of HNTs on the swelling ratios of the polysaccharide/HNTs composites were investigated in NaCl and water solution. Compared with pure sodium alginate (SA) hydrogel, the SA/HNTs composite hydrogels showed low swelling ratios with the same conditions for soaking time, which gradually decreased with the increasing HNTs loading. This result was attributed to the hydrophilic polymer content in the composite hydrogels decreasing with the addition of HNTs, and the water adsorption of HNTs was lower than SA. In addition, the HNTs used as physical crosslinking points for alginate through the hydrogen bond interactions can greatly improve entanglement of the alginate and lower the mobility of the chains, resulting in water absorption being greatly decreased [47]. Sinem et al. reported a cryogenic technique to modify HNTs. The inner and outer diameters and the surface area of HNTs were evidently increased without disturbing the inherent tubular structure and wall features. Then, modified HNTs were mixed with chitosan to prepared composite hydrogels, showing remarkedly improved mechanical and swelling properties compared with pure chitosan hydrogel [48]. Sharifzadeh et al. synthesized carrageenan/HNTs nanocomposite hydrogels via physical crosslinking. The chemical structure confirmed by FTIR spectroscopy revealed the formation of physical interaction between carrageenan and HNTs in the hydrogels. It was revealed that the thermal stability and swelling of the nanocomposite hydrogels had significantly been improved due to the incorporation of HNTs compared with the pure carrageenan hydrogel [49]. The reasons why HNTs can improve the thermal stability of composites are as follows. The degradation temperature of HNTs is approximately 400 °C, which is higher than most of the polysaccharides. Then, the dispersed HNTs have a blocking effect on mass and heat transfer. Besides, the polysaccharide chains and degraded products enter the inner cavity of HNTs, delaying mass transport and improving the thermal stability. However, the good dispersion of HNTs into the hydrogel is urgently needed for the hydrogel fabrications to broaden their application. The HNTs functionalized via different types of silane coupling agents were used as a way to improve HNTs dispersal in the polymer matrix. Sabbagh et al. prepared novel chitosan/crosslinked oxidized starch hydrogels, which were embedded by modified or unmodified HNTs. Incorporation of HNTs significantly affected the swelling behavior and thermal properties of the hydrogel. The increase of the amine groups in HNTs modified with silane reagents made them react with oxidized starch, resulting in good dispersion in the structure of the hydrogel [50]. Figure 3 illustrates the formation of the bio-nanocomposite hydrogel.

The swelling ratio of the chitosan/HNTs hydrogel also decreased compared with the pure chitosan hydrogel, due to the introduction of HNTs content causing the chitosan to contract more [51]. The cellulose/HNTs composite showed a similar variation trend [52].

#### 2.2.2. Films

Regenerated cellulose/HNTs nanocomposite films were fabricated in 1-butyl-3-methylimidazolium chloride ionic liquid by solution casting method. Figure 4 showed the cross-sectional FE-SEM images of the cellulose and 6 wt.% HNTs-filled nanocomposite films. The HNTs were well dispersed in cellulose due to good interaction between cellulose and HNTs. Young’s modulus and the tensile strength of nanocomposite films were improved by 100% and 55.3%, respectively, when the loading of HNTs was 6 wt.%, which was owing to tubular geometry and the higher stiffness of the HNTs. The addition of HNTs also improved the thermal stability and char yield of regenerated cellulose, but moisture absorption capacity of the nanocomposites in constant relative humidity was reduced due to the addition of HNTs [53].

Kim et al. reported transparent cellulose-obtained films from cellulose/HNTs solutions. The HNTs could uniformly been dispersed in cellulose because of the repulsive force from its surface charge, and the hydrogen bonding from HNTs and cellulose broke the chain-to-chain interactions of cellulose. The haze of the film was increased due to the introduction of HNTs but the diffuse transmittance could be retained [54].

Chang et al. prepared dispersed starch/HNTs composite films by using amylose to wrap the HNTs by ball-milling [55], in order to solve the agglomeration of HNTs. However, the extraction of amylose is expensive and complicated. In one work, polyethylene glycol (PEG) was used as a dispersing agent to mill, modify, and disperse HNTs in different solvents, and certain amounts of glycerin and modified HNTs suspension were added into the slurry. The composite films were obtained after stirring and casting on the stainless-steel plate. SEM of treated HNTs and HNTs/starch films with 3 wt.% (c and d) and 7 wt.% HNTs were shown in Figure 5. The HNTs are evenly distributed in the starch matrix. Due to the action of PEG, the treated HNTs were well dispersed in the starch matrix and the tensile strength of the film was effectively improved [56,57]. In another work, chitosan/starch/HNTs ternary nanocomposite films were developed through solution casting method. The interactions between chitosan, starch, and halloysite nanotubes were confirmed by FTIR results. Water absorption capacity, folding strength, and hemocompatibility were remarkedly enhanced owing to the addition of halloysite nanotubes [58]. Then, chitosan-HNTs composites were combined with modified cellulose to produce composite films using a solution casting method [59]; the excellent film formation and increase in surface roughness of the nanocomposite were confirmed by morphological and surface analysis. The kappa carrageenan/HNTs bio-nanocomposite films with enhanced tensile properties were successfully fabricated [60].

A functional bio-nanocomposite film both with antioxidant and antimicrobial active molecules was successfully prepared by the filling of a pectin matrix with modified HNTs containing peppermint oil. Importantly, the prepared functional film was considered a biocompatible material for packaging applications because of it was composed of eco-compatible molecules [61]. Makaremi et al. developed functional films with antimicrobial properties that can be extended over time by dispersing a HNTs/salicylic acid hybrid into the pectin matrix [62]. Moreover, it was demonstrated that the vacuum pumping in/out procedure can optimize the halloysite loading from the aqueous solution because of the water confinement mechanism [63]. Accordingly, the attained knowledge can offer new routes in the preparation of effective delivery systems based on HNTs.

#### 2.2.3. Fibers

Fibers based on polysaccharides with high performance or special functions are becoming an emerging hotspot in both academic and industrial circles, exhibiting great potential in multiple fields.

HNTs can be well-dispersed in some ionic liquids. Song et al. firstly reported fabrication of microcrystalline cellulose/HNTs composite fibers from cellulose/ionic liquid/HNTs solutions by a wet-spinning method. Figure 6 showed schematic apparatus of nanocomposite fibers and SEM images of the morphology of fibers. The uniform dispersion of HNTs in the cellulose matrix was further confirmed by FTIR and wide-angle X-ray diffraction (WAXD) spectra. In addition, the homogeneous dispersion of HNTs and strong interfacial adhesion between HNTs and cellulose chains dramatically enhanced the moisture barrier property of cellulose fibers. Mechanical and thermal properties of HNTs nanocomposite fibers were greatly enhanced due to the uniform orientation of the well dispersed HNTs and great interactions from cellulose and HNTs caused by hydrogen bonding, van der Waals, and electrostatic interactions. For example, the tensile strength of the fibers increases from 73.8 to 130.1 MPa with the addition of HNTs from 0 to 7 wt.% [64].

Silva et al. developed HNTs reinforced alginate nanofibrous scaffolds fabricated by electrospinning. The diameter of alginate-based nanofibers ranged from 40 to 522 nm with well-aligned HNTs, as shown in Figure 7. The HNTs were well dispersed in the alginate matrix with good uniaxial alignment. Compared to the alginate-based scaffolds without HNTs, the tensile strength and elastic modulus of HNTs-reinforced nanofibrous scaffolds were significantly improved by 3-fold and 2-fold, respectively, when 5% (*w*/*w*) HNTs was added. The incorporation of HNTs also enhanced the thermal stability of the nanofibrous scaffolds [65].

#### 2.2.4. Plasticized Nanocomposites

Plasticized halloysite nanocomposites are commonly prepared via solvent casting method, but this is limited in the practical processing industries. Thus, the melt mixing technique will be more efficient and productive in a process industry workshop [66].

Schmitt et al. successfully developed plasticized HNTs/wheat starch nanocomposites by melt-extrusion for the first time. A higher loading (up to 6 wt.%) of HNTs were well-dispersed in a starch matrix. The interactions between external hydroxyl groups of halloysites and C-O-C groups of starch were formed. The thermal stability of the matrix was improved with the addition of HNTs. The tensile strength and Young’s modulus of starch were improved up to 29% and 144%, respectively, without sacrificing ductility [67]. Porous plasticized starch/HNTs nanocomposites were also successfully prepared by melt-extrusion technique. Double benefits could be gained due to the addition of HNTs, which were as a nucleating agent increasing the porosity and as a barrier agent increasing the proportion of small cells [68]. Ren et al. reported plasticized starch/HNTs nanocomposites prepared by melt blending with different polyol plasticizers, such as glycerol, sorbitol, and a mixture of two. Compared to sorbitol or a mixture of two, glycerol offered a more uniform dispersion of HNTs in the starch matrix owing to more stable hydrogen bonds from glycerol and HNTs. The use of mixtures of these polyols was proved a promising way to optimize the mechanical properties of nanocomposites [69].

## 3. Applications of Polysaccharide-HNTs Composites

### 3.1. Biomedical Applications

Liu et al. summarized recent research progress in the biomedical application of polysaccharide-HNTs composites [39]. The interfacial interactions, structure, and properties of the composites were discussed in detail in that review. There were some examples in the biomedical applications, such as tissue engineering, wound healing, and drug carrier systems. Thus, we have listed some examples based on the previous work of Liu et al. in the following.

#### 3.1.1. Drug Delivery and Release

HNTs were confirmed as potential drug and gene delivery vehicles [70,71], and the drug release rate slowed down by coating polysaccharides onto the drug-loaded HNTs. Recently, more and more attention has been paid to the drug sustained release based on the polysaccharide-HNTs composites [72,73].

Hydrogels have been researched for the controlled delivery of biomolecules, varying from small molecular weight drugs to biomacromolecules, such as nucleic acids, polysaccharides, and proteins. Moreover, the biocompatible and biodegradable hydrogels were prepared by different natural ingredients. Among them, the chitosan-HNTs composite hydrogels were widely researched due to their low toxicity, good biocompatibility, and degradability by human enzymes [74]. It was found that the chitosan-coated HNTs exhibited reduced release compared with the pure HNTs [24]. For instance, the chitosan-coated HNTs had released only 78% of the total drug payload, while the uncoated HNTs released 88% on day 9. The drug release rate was extremely low, and the residual content was under 10% of the loaded drug after 20 days. The reason why chitosan-coated HNTs have a lower drug release rate is the additional barrier through which the drug must diffuse provided by chitosan. In practical applications, chitosan-based hydrogels have been used for cancer therapeutics, subcutaneous release, and oral delivery. In recent years, the relevant relationship between plasticizer nature and drug release behavior has been exploited, so the starch/HNTs composite films have been prepared by melt blending technique for drug release applications [75]. In addition to drugs, other active agents, such as antimicrobial agents [76], DNA [77], and proteins, can also be loaded for controlled release.

#### 3.1.2. Tissue Engineering Scaffold

Tissue engineering scaffolds are generally porous structures, biocompatible, and mechanically strong for shaping cell growth [78]. HNTs were added into polysaccharides, such as chitosan [79], alginate [47], and starch [68], to prepare various tissue engineering scaffolds. Compared with pristine chitosan, the chitosan-HNTs nanocomposite scaffolds showed an obvious improvement in mechanical strength, tensile modulus, and thermal stability. Moreover, the addition of HNTs had little influence on the pore structure of chitosan. Thus, the chitosan-HNTs exhibited a highly porous structure. In order to verify the feasibility of the nanocomposite scaffolds, mouse fibroblasts were used for culture. The result showed that mouse fibroblasts could develop on the chitosan-HNTs nanocomposite scaffold surfaces, even at 80 wt.% HNTs loading. In addition, the fibroblast cells cultured on the surfaces exhibited a phenotypic shape, indicating that the cells could penetrate and migrate within the scaffolds, which were similar to the extracellular matrix. Thus, polysaccharide-HNTs composites have promising potential in the tissue engineering scaffold field.

#### 3.1.3. Wound Dressing

Wound dressings can be classified as traditional, biomaterial-based, interactive, and bioactive dressings. In recent research studies, biomaterials, such as polysaccharides, which are non-toxic, natural available, non-immunogenic, biocompatible, and biodegradable, have been considered as ideal materials for wound healing [80,81,82]. Due to the characteristics of biocompatibility, ability to load and release bioactive agents, high water content, and flexibility, hydrogels are most widely adopted for wound dressings [83]. Moreover, HNTs have been added into the polysaccharides, such as alginate [45,76] and chitosan [84,85], to overcome the poor mechanical strength of hydrogels. The alginate-HNTs wound dressing, based on a double barrier with the antibiotic vancomycin as an antimicrobial agent, was prepared. It was concluded that only the antibiotic vancomycin immobilized in HNTs- (3-Aminopropyl)-trimethoxysilane (APTS) and encapsulated in alginate hydrogels can be used as a wound dressing material.

### 3.2. Wastewater Treatment Applications

Among all the causes of wastewater pollutants, heavy metal ions, dyes, and other organic pollutants account for a large proportion [86]. Various methods, such as flocculation [87], precipitation [88], membrane filtration [89], and electrochemical techniques [90] have been restricted in practical usage because of the high cost, secondary pollutants, and poor removal efficiency. Due to its low cost, simple operation, and potential recycling and reuse, adsorption has become one of the most effective alternatives. Traditional absorbents, such as activated carbon [91] and resins [92], have been replaced by HNTs, a novel absorbent which is inexpensive and naturally abundant. In recent research studies, polysaccharide-HNTs composites exhibited promising adsorption capacity and regeneration.

#### 3.2.1. Applications in Removal of Dyes

Pristine HNTs with the negative Si-O-Si on the outer surface and the positive Al-OH on the inner surface can adsorb both cationic and anionic dyes, but stable dispersions in water and reusability for its practical application are hard to obtain due to their size. Polysaccharides, such as chitosan, alginate, and starch, have been classified as natural biopolymer adsorbents, as they are non-toxic and biodegradable. However, polysaccharides materials usually have dense layer, low mechanical strength, and rigidity characteristics, which limits their application on a practical scale. Thus, polysaccharide-HNTs composites have been attractive due to combining their benefits for treatment of wastewater.

Cavallaro et al. prepared alginate/HNTs composite hydrogels by encapsulating HNTs into alginate hydrogels using the dropping technique. The adsorption capacity of composite hydrogels was investigated by removing crystal violet (CV) from aqueous media. The results showed that the addition of HNTs enhanced the composite hydrogel’s ability to capture CV. The weight ratio of alginate and HNTs was 2:1, and the removal rate was increased by 55% and 45%, respectively, when the stoichiometric adsorbent concentration increased from 0.25 to 0.50 mass% [93]. It was found that the pH and temperature had little influence on the adsorption capacity of alginate-HNTs composite beads. The alginate-HNTs composite adsorbents (10 g) had the ability to treat 29.7 L of 55 mg/L methylene blue (MB) solution and the removal efficiency was above 90%. Moreover, the column study verified that the removal efficiency had a slight decline with the increase of bed volumes, but it remained above 90% after 1500 bed volumes of wastewater was treated [94]. Novel chitosan-HNTs composite hydrogel beads were prepared by the dropping and pH-precipitation technique, exhibiting accelerated adsorption process and improved adsorption capacity (72.60 mg/g for MB and 276.9 mg/g for malachite green (MG)) compared with the pure chitosan bead. Similar to alginate-HNTs composites, chitosan-HNTs composites also exhibited excellent regeneration properties especially for MB (above 92%). The removal ratio of dyes increased but the adsorption amount per unit absorbent weight reduced with the addition of the composite hydrogels in solution [95]. The porous starch-HNTs composites were prepared by solvent exchange method using different drying methods. The adsorption performance depended on porosity, which was relative to specific surface area, and the size distribution of the pores and their area changed with different types of drying methods and percentage of ethanol. The adsorption capacity composites were improved due to the incorporation of HNTs [96]. A cryogenic technique to modify HNTs was confirmed to increase the inner and outer diameters and the surface area of HNTs without disturbing the inherent tubular structure and wall features. A small amount of cryo-expanded halloysite was used in chitosan. It was found that the composite hydrogel had a high adsorption capacity for anionic and cationic dyes [48].

Overall, polysaccharide-HNTs composites exhibit reinforced adsorption capacity and regeneration properties. Thus, polysaccharide-HNTs composites can be considered as promising reusable adsorbents for the removal of dyes from wastewater.

#### 3.2.2. Applications in Removal of Heavy Metal Ions

HNTs with negatively charged surfaces were regarded as ideal alternatives for removal of heavy metal ions from aqueous media, such as Cu^2+^, Pb^2+^, and Cr^2+^ [14,97]. However, HNTs used as adsorbents face some problems in practical application due to their nanoscale size, such as being easily aggregated, being lost in use, the low permeability of packed adsorption columns, and being difficult to recycle. Thus, HNTs were combined with polysaccharides to improve the affinity and loading capacity for heavy metal ions. Extrusion dripping method was used to produce alginate-HNTs nanocomposite beads with calcium chloride as the curing agent. The results showed that the alginate-HNTs nanocomposite beads had improved adsorption capacity (325 mg/g at 0.2 g HNTs loading). It was also concluded that alginate and HNTs promoted the adsorption of Pb^2+^ by ion exchange and physisorption, respectively. It was simple to separate the used nanocomposite beads from wastewater by filtration due to the millimetric size, which facilitated the regeneration of the used beads [98]. In other works, the alginate-HNTs hybrid beads with high adsorption capacity for Cu^2+^ were prepared in the same way. The SEM image showed clearly that the HNTs overlapped loosely together in the interior of the hybrid beads, contributing to the mass transfer of the adsorbents. Furthermore, the regeneration experiment exhibited good adsorption efficiency (approximately 80%) after three cycles. In addition, the alginate-HNTs hybrid beads also showed good adsorption performance for other heavy metal ions, such as Ag^+^ and Cd^2+^ [99].

Compared to the traditional chemical methods of removing heavy metal ions from aqueous media, polysaccharide-HNTs composites had advantages, such as low cost, simple operation, regeneration, and being non-secondary pollutants.

### 3.3. Other Applications for Water Treatment

Currently, the research is focused on the applications of polysaccharide-HNTs composites for the removal of dyes and heavy metal ions in aqueous media. However, polysaccharide-HNTs composites have other practical applications for pollutants removal due to their unique structure and improved properties.

Different types of membranes, such as ultrafiltration, forward osmosis, reverse osmosis, and membrane reactors [100,101,102,103], were endowed with special performances by adopting HNTs. Cellulose acetate/L-dopa coated HNTs (LDPHNT) ultrafiltration membranes were prepared by blending in a casting solution. The Energy dispersive spectroscopy (EDS) analysis showed the uniform dispersion of LPDHNTs in the cellulose acetate (CA) membrane matrix. Moreover, the size and number of pores on the membrane surface were increased by the addition of LDPHNTs into CA solution. It was the existence of LDPHNTs that improved the hydrophilicity of the hybrid membranes. In the antifouling test, the composite membranes exhibited higher antifouling performance than pristine CA membranes. Besides, the tensile modulus and elongation at break point of LDPHNTs-CA hybrid membranes were reinforced compared to the pure CA membranes [104]. Some of the phenol-containing pesticides are widely used in agriculture and regarded as endocrine disruptors, exhibiting estrogenic activity and toxicity even at low concentrations. In order to remove phenol-containing pollutants, the special adsorbents based on HNTs were prepared. The chitosan-HNTs hybrid nanotubes were synthesized by simply assembling chitosan onto HNTs. Then, the horseradish peroxidase was immobilized on the hybrid nanotubes for the removal of phenol. The experiment showed that the immobilized HRP exhibited excellent removal efficiency for phenol from wastewater and the activity did not reduce the volume [25].

### 3.4. Food Packaging Applications

Over the years, polysaccharides, owing to their biodegradability, non-toxicity, and good film-forming ability, have raised concerns over their use as food packaging materials for consumer demand and environmental issues. However, these biodegradable polysaccharides have weak stability in processing, poor barrier properties, and high sensitivity to environmental changes. In the previous work, the introduction of HNTs nanoparticles to the starch matrix improved the mechanical properties and decreased permeability to water vapor and oxygen, water adsorption capacity, as well as the water solubility of the films [57], which gave them potential to be used for food packaging purposes. Makaremi et al. prepared biofilms composed of apple pectin and two different types of HNTs—MB with shorter tubes and lower surface area and PT with longer tubes and higher surface area—to obtain a novel functional bio-nanocomposite with enhanced mechanical and thermal properties. Moreover, both HNTs were employed as nanocontainers for salicylic acid, a well-known biocidal agent. On this basis, the HNTs/salicylic acid hybrids were dispersed into the apple pectin matrix to develop functional films with antimicrobial properties that can be extended over time. Thus, the bio-nanocomposite films showed promising potential for food packaging applications [62].

HNTs are able to adsorb active molecules, such as nisin and pediocin. On this basis, Meira et al. added peptides nisin and pediocin into starch films, resulting in active packaging materials with antimicrobial activity against *L. monocytogenes* and *C. perfringens*. The addition of HNTs enhanced the mechanical and thermal properties, especially when bacteriocins were adsorbed on the HNTs [105]. The moisture barrier properties of polysaccharide-based films are poor due to their hydrophilic characteristics. Essential oils with antioxidant and antimicrobial activities are commonly incorporated into polysaccharide matrices to overcome these limitations. However, it is rather hard to disperse the essential oils in a hydrophilic polysaccharide matrix. Lee et al. developed chitosan films incorporated with clove essential oil and HNTs. It was confirmed that the essential oils were stabilized by HNTs without any surfactant. The addition of HNTs into the chitosan matrix enhanced the mechanical and water barrier properties of the chitosan films, and the active molecule, clove essential oil, imparted antimicrobial and antioxidant effects to the chitosan-HNTs nanocomposite films. The results showed that the films inhibited growth of mold, which was derived from the surrounding environment. Therefore, the nanocomposite films could be used as active food packaging systems because of the antioxidant and antimicrobial properties and enhanced barrier properties against water vapor [106].

## 4. Conclusions

In this review, we summarized the recent research studies regarding polysaccharide-HNT composites. The key points of polysaccharide-HNTs composites are as follows:(1)HNTs, 1D natural nanoclays, have unique characteristics of tubular structure, high aspect ratio, abundant natural reserves, compatibility, and high mechanical strength. Due to the characteristics of HNTs, polysaccharide-HNTs composites have advantages, such as improved mechanical, thermal, and swelling properties and good biocompatibility. Thus, HNTs are promising nanofillers for high-performance polymer composites.(2)In addition to the characteristics of HNTs, the degree of dispersion of HNTs and the interfacial interactions between polysaccharides and HNTs (electrostatic and hydrogen bonding interactions) are crucial factors affecting the performance of composites.(3)HNTs can be combined with polysaccharides by different methods. Polysaccharide-HNTs composite hydrogels can be prepared by solution mixing and freeze-drying method and dropping or PH-precipitation technique; membranes are fabricated by solution casting method and fibers are usually produced by electrospinning technique. The key to this process is to obtain a well-dispersed solution with the HNTs and good interfacial interactions between polysaccharides and HNTs.(4)Polysaccharide-HNTs composites show promising potential for biomedical applications. The applications in removal of dyes and heavy metal ions are summarized in detail. Polysaccharide-HNTs composites have raised concerns as food packaging materials for consumer demand and environmental issues.

## Figures and Tables

**Figure 1 polymers-11-00987-f001:**
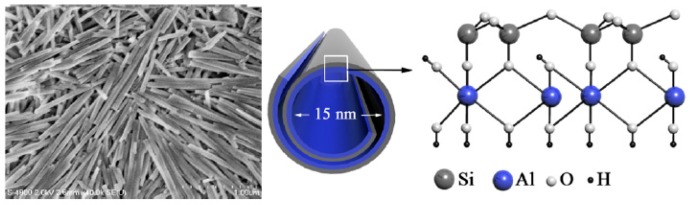
FE-SEM Image of HNTs on Si-Wafer (Left) and Schematic Illustration of Crystalline Structure of HNTs (Right). (Reproduced from [42] with permission from American Chemical Society and Copyright Clearance Center, 2012).

**Figure 2 polymers-11-00987-f002:**
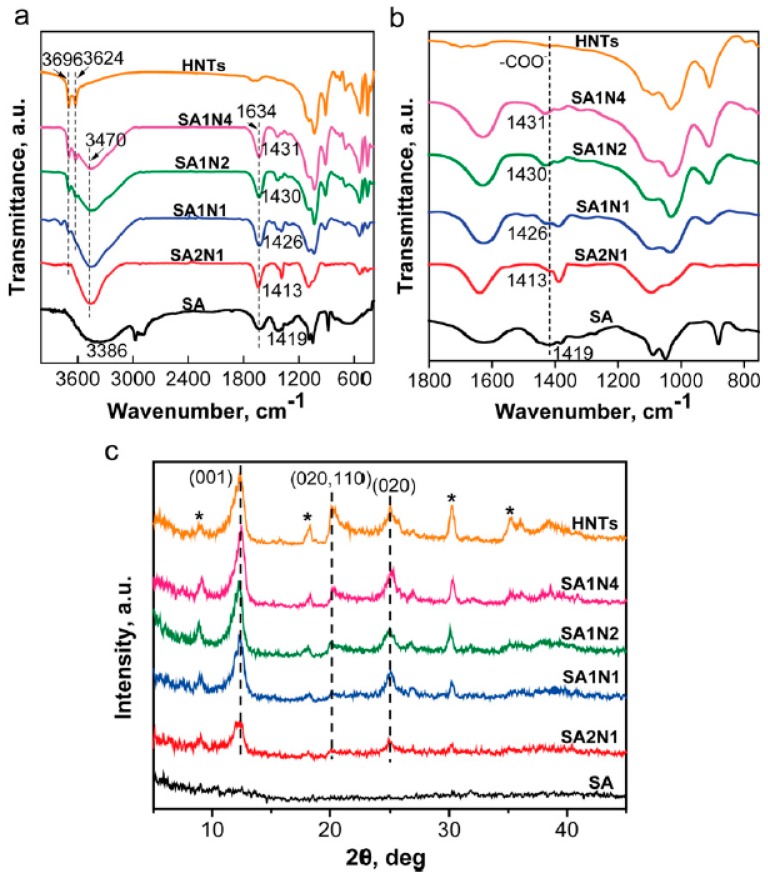
FTIR spectra (**a**,**b**) and XRD (**c**) pattern of HNTs, alginate, and alginate/HNTs composites. (Reproduced from [46] with permission from Elsevier and Copyright Clearance Center, 2017).

**Figure 3 polymers-11-00987-f003:**
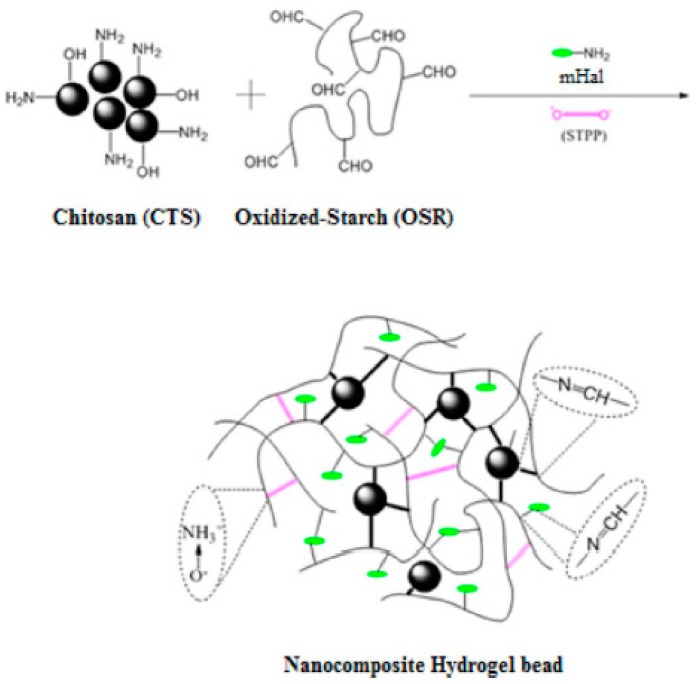
Schematic description of bio-nanocomposite hydrogel formation. (Reproduced from [50] with permission from Elsevier and Copyright Clearance Center, 2017).

**Figure 4 polymers-11-00987-f004:**
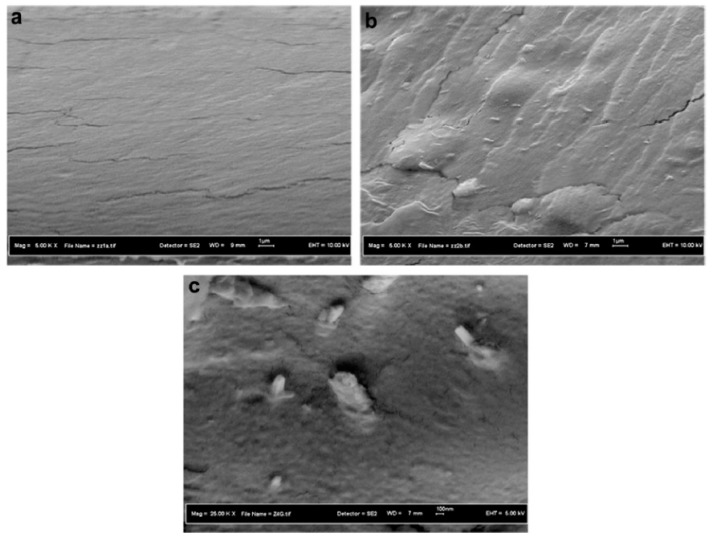
Cross-sectional FE-SEM images of cellulose and 6 wt.% HNTs filled cellulose/HNTs nanocomposite films under (**a**,**b**) low magnifications and (**c**) high magnification. (Reproduced from Reference [53] with permission from Elsevier and Copyright Clearance Center, 2013).

**Figure 5 polymers-11-00987-f005:**
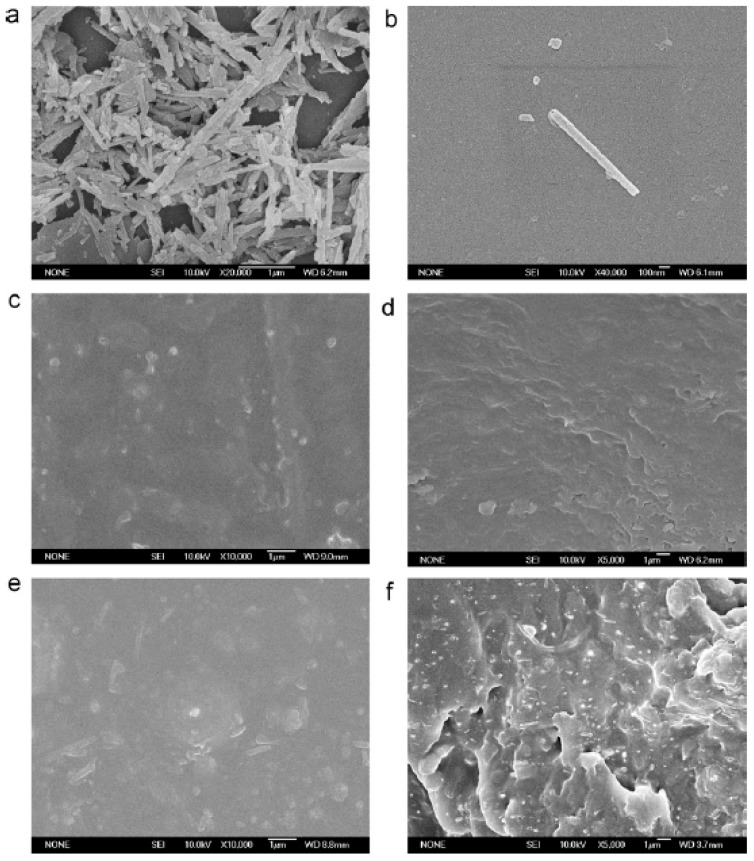
SEM of treated halloysite (**a**,**b**) and halloysite/starch films with 3 wt.% (**c**,**d**) and 7 wt.% halloysite (**e**,**f**). (Reproduced from [56] with permission from Elsevier and Copyright Clearance Center, 2012).

**Figure 6 polymers-11-00987-f006:**
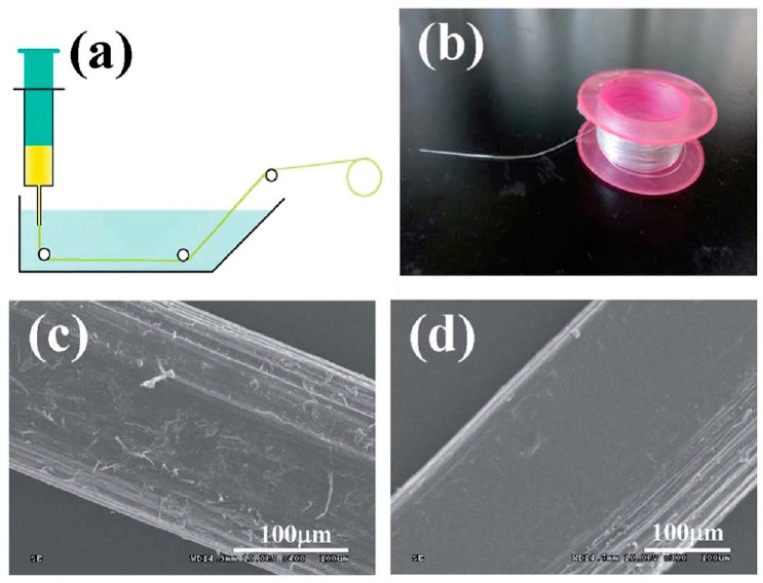
(**a**) Schematic apparatus for spinning MCC/HNTs nanocomposite fibers. (**b**) A 3 m long MCC/HNTs fiber wound on a plastic reel. (**c**,**d**) SEM images of the morphology of the side surface sections of regenerated MCC and MCC/HNTs fibers. (Reproduced from [64] with permission from Royal Society of Chemistry and Copyright Clearance Center, 2014).

**Figure 7 polymers-11-00987-f007:**
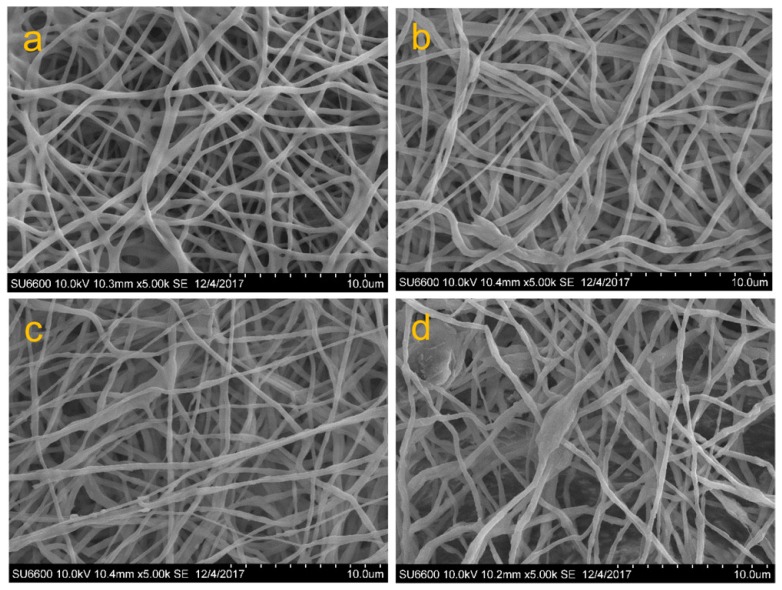
Morphology of electrospun alginate-based scaffolds with (**a**) 0, (**b**,**c**) 5%, and (**d**) 10% (*w*/*w*) of HNTs. (Reproduced from [65] with permission from American Chemical Society and Copyright Clearance Center, 2018).

**Table 1 polymers-11-00987-t001:** The detailed data of halloysite nanotubes (HNTs) related to combination with polysaccharides.

Molecular Formula	Al_2_Si_2_O_5_(OH)_4_·nH_2_O
Length	100–2000 nm
Inner diameter	10–30 nm
Outer diameter	30–50 nm
Aspect ratio (L/D)	10–50
Young’s modulus of a single HNTs	130 ± 24 GPa
Elastic modulus	460 GPa
Interlayer water removal temperature	400 °C
Water contact angle	10 ± 3°
Specific surface area	22.1–81.6 m^2^/g
Total pore volume	0.06–0.25 cm^3^/g
Density	2.14–2.59 g/cm^3^
Mean particle size in aqueous solution	143 nm

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
