# Peer review of "Advances in Halloysite Nanotubes–Polysaccharide Nanocomposite Preparation and Applications"

_polymers, 2019, doi:10.3390/polym11060987_

Reviewer 1 Report

The manuscript is a review, dealing with the recent progress in the field of  polysaccharide/Halloysite nanotubes composites and their application in biomedical and wastewater field. The choice fulfills the request of the editor of covering specific material/field of application. The topic is interesting and the cutting edge, as can be seen by the many articles published in the field in the last years. Liu et al. (Clay Minerals, Volume 51, Issue 3, Pages 457–467, ISSN (Online) 1471-8030, ISSN (Print) 0009-8558, DOI: https://doi.org/10.1180/claymin.2016.051.3.02.) published a review on the similar topic but focused on biomedical application. The author should cite and underline the differences with this review to avoid superposition and modify/reduce section 3.1 accordingly. At the same time I suggest adding a 3.4 section about the use of polysaccharide/Halloysite nanotubes in for food packaging as in Makaremi et al., ACS Appl. Mater. Interfaces, 2017, 9 (20), pp 17476–17488 for instance.
Minor notes
1)    Line 30: “novel”: HNT are not so novel, I suggest to eliminate or change or specify differently their “age”
2)    Line 75: “2.2. The main existence forms” This title is not clear and misleading. Please change it “HNT/polysaccaride preparations and formulations” for instance?
3)    Line 82 “hydroscopicity “: I guess this is a typo, please see https://www.dictionary.com/browse/hydroscopicity”
4)    Line 110 “got well close” can be “resulted very similar”?
5)    Line 156 “retained” instead of “kept”?
6)    Line 188-189  “greatly interactions from cellulose and HNTs by hydrogen bonding, van der Waals and electrostatic interactions” This sentence is not connected to the rest sentence a verb seems missing: to be rewritten
7)    Line 207-208 “but which is 207 limited” to be rewritten, it is not correct English
8)    Line 225: As written above, the section 3.1 should not be coincident with Liu et al. (Clay Minerals, Volume 51, Issue 3, Pages 457–467 cited above and the differences must be stated at the beginning of the section
9)    I suggest adding a section 3.4 for application of HNT-polysaccarides in polymer for food packaging, another promising field of application of HNT-polysaccarides
10)    Line 404: add a comment on food packaging applications

Author Response

Response to Reviewer 1 Comments

We are grateful for your comments and criticism. Indeed, your comments are very helpful for refining the original manuscript to a revised one. We took the following actions to your comments. We hope if these changes would be sufficient for publication of this paper in Polymers.

Liu et al. (Clay Minerals, Volume 51, Issue 3, Pages 457–467, ISSN (Online) 1471-8030, ISSN (Print) 0009-8558, DOI: https://doi.org/10.1180/claymin.2016.051.3.02.) published a review on the similar topic but focused on biomedical application. The author should cite and underline the differences with this review to avoid superposition and modify/reduce section 3.1 accordingly.

Reply: Many thanks for the reviewer’s suggestion. Although general properties of polysaccharide/halloysite nanotube composites and biomedical applications have been reviewed earlier by Liu et al [37], we review the recent progress toward the development of polysaccharide-HNTs composites, paying attention to the main existence forms, wastewater treatment and food packaging applications particularly. Through this review, we can have a better understanding of unique characteristics of polysaccharide-HNTs composites, which can be helpful to the continuous expansion of their application fields in the future. (Lines79-85, Page 3, in the revised manuscript)

At the same time I suggest adding a 3.4 section about the use of polysaccharide/Halloysite nanotubes in for food packaging as in Makaremi et al., ACS Appl. Mater. Interfaces, 2017, 9 (20), pp 17476–17488 for instance.

 Reply: Many thanks for the reviewer’s suggestion. Over the years, polysaccharides, owing to their biodegradability, non-toxicity, good film-forming ability, have been raised concerns as food packaging materials for consumer demand and environmental issues. However, these biodegradable polysaccharides are weak stability in processing, poor barrier properties, and high sensitivity to environmental changes. In the previous work, the introduction of HNTs nanoparticles to starch matrix improved mechanical property and decreased permeability to water vapor and oxygen, water adsorption capacity as well as water solubility of the films [57], which was potential to be used for food packaging purposes. Makaremi et al. prepared biofilms composed of apple pectin and two different types of HNTs, MB with shorter tubes and lower surface area and PT with longer tubes and higher surface area, to obtain a novel functional bionanocomposite with enhanced mechanical and thermal properties. Moreover, both HNTs were employed as nanocontainers for salicylic acid, a well-known biocidal agent. On the basis, the HNTs/salicylic acid hybrids were dispersed into the apple pectin matrix to develop functional films with antimicrobial properties which can be extended over time. Thus, the bionanocomposite films showed promising potential for food packaging applications [62]. HNTs are able to adsorb active molecules, such as nisin and pediocin. On the basis, Meira et al. added peptides nisin and pediocin into starch films, resulting in active packaging materials with antimicrobial activity against L. monocytogenes and C. perfringens. The addition of HNTs enhanced mechanical and thermal properties, especially when bacteriocins were adsorbed on the HNTs [105]. The moisture barrier properties polysaccharides-based films are poor due to their hydrophilic characteristics. Essentialoils with antioxidant and antimicrobial activities are commonly incorporated into polysaccharides matrix to overcome these limitations. However, it is rather hard to disperse the essential oil in the hydrophilic polysaccharides matrix. Lee et al. developed chitosan films incorporated with clove essential oil and HNTs. It was confirmed that the essential oils would be stabilized by HNTs without any surfactant. The addition of HNTs into the chitosan matrix enhanced the mechanical and water barrier properties of the chitosan films, and the active molecule, clove essential oil, imparted antimicrobial and antioxidant effects to the chitosan-HNTs nanocomposite films. The results showed that the films inhibited growth of mold which was derived from the surrounding environment. Therefore, the nanocomposite films could be used as active food packaging systems because of the antioxidant, antimicrobial properties and enhanced barrier property against water vapor [106]. (Lines452-488, Page 17-18, in the revised manuscript)

Minor notes

1) Line 30: “novel”: HNT are not so novel, I suggest to eliminate or change or specify differently their “age”.

 Reply: Many thanks for the reviewer’s suggestion. We have eliminated the word “novel” in the revised manuscript.

2) Line 75: “2.2. The main existence forms” This title is not clear and misleading. Please change it “HNT/polysaccharide preparations and formulations” for instance?

Reply: Many thanks for the reviewer’s comment. The title of “The main existence forms” has been changed by “HNT/polysaccharide preparations and formulations” in the revised manuscript. (Lines102, Page 4, in the revised manuscript)

3) Line 82 “hydroscopicity”: I guess this is a typo, please see https://www.dictionary.com/browse/hydroscopicity.

Reply: Thanks for the reviewer’s good question. This is indeed a spelling mistake. We have corrected “hydroscopicity” to “hygroscopicity” in the revised manuscript. (Lines110, Page 4, in the revised manuscript)

4) Line 110: “got well close” can be “resulted very similar”?

Reply: Thanks for the reviewer’s question. We have modified “got well close” to “resulted very similar” in the revised manuscript. (Lines132, Page 5, in the revised manuscript)

5) Line 156 “retained” instead of “kept”?

Reply: Many thanks for the reviewer’s comment. We have changed the word “kept” to “retained” in the revised manuscript. (Lines200, Page 9, in the revised manuscript)

6) Line 188-189: “greatly interactions from cellulose and HNTs by hydrogen bonding, van der Waals and electrostatic interactions” This sentence is not connected to the rest sentence a verb seems missing: to be rewritten

Reply: Many thanks for the reviewer’s comment. We have rewritten the sentence in the revised manuscript: “greatly interactions from cellulose and HNTs caused by hydrogen bonding, van der Waals and electrostatic interactions”. (Lines248-249 Page 10, in the revised manuscript)

7) Line 207-208 “but which is 207 limited” to be rewritten, it is not correct English.

Reply: Many thanks for the reviewer’s comment. We have rewritten the sentence in the revised manuscript: “Plasticized halloysite nanocomposites are commonly prepared via solvent casting method, but it is limited in the practical processing industries”. (Lines270-271, Page 12, in the revised manuscript)

8) Line 225: As written above, the section 3.1 should not be coincident with Liu et al. (Clay Minerals, Volume 51, Issue 3, Pages 457–467 cited above and the differences must be stated at the beginning of the section.

Reply: Thanks for the reviewer’s suggestion. We have added the differences at the beginning of the section and modified some content appropriately in the revised manuscript. Liu et al. summarized recent research progress in the biomedical application of polysaccharide-HNT composites [37]. The interfacial interactions, structure and properties of the composites were discussed in detail in that review. There were few examples in the biomedical applications such as tissue engineering, wound healing and drug carrier system. Thus, we have listed some examples based on the previous work of Liu et al. in the following. (Lines291-296, Page 13, in the revised manuscript)

9) I suggest adding a section 3.4 for application of HNT-polysaccharide in polymer for food packaging, another promising field of application of HNT-polysaccharides.

Reply: Thanks for the reviewer’s advice. We have added a section 3.4 for application of HNT-polysaccharide in polymer for food packaging in the revised manuscript. Over the years, polysaccharides, owing to their biodegradability, non-toxicity, good film-forming ability, have been raised concerns as food packaging materials for consumer demand and environmental issues. However, these biodegradable polysaccharides are weak stability in processing, poor barrier properties, and high sensitivity to environmental changes. In the previous work, the introduction of HNTs nanoparticles to starch matrix improved mechanical property and decreased permeability to water vapor and oxygen, water adsorption capacity as well as water solubility of the films [57], which was potential to be used for food packaging purposes. Makaremi et al. prepared biofilms composed of apple pectin and two different types of HNTs, MB with shorter tubes and lower surface area and PT with longer tubes and higher surface area, to obtain a novel functional bionanocomposite with enhanced mechanical and thermal properties. Moreover, both HNTs were employed as nanocontainers for salicylic acid, a well-known biocidal agent. On the basis, the HNTs/salicylic acid hybrids were dispersed into the apple pectin matrix to develop functional films with antimicrobial properties which can be extended over time. Thus, the bionanocomposite films showed promising potential for food packaging applications [62]. HNTs are able to adsorb active molecules, such as nisin and pediocin. On the basis, Meira et al. added peptides nisin and pediocin into starch films, resulting in active packaging materials with antimicrobial activity against L. monocytogenes and C. perfringens. The addition of HNTs enhanced mechanical and thermal properties, especially when bacteriocins were adsorbed on the HNTs [105]. The moisture barrier properties polysaccharides-based films are poor due to their hydrophilic characteristics. Essentialoils with antioxidant and antimicrobial activities are commonly incorporated into polysaccharides matrix to overcome these limitations. However, it is rather hard to disperse the essential oil in the hydrophilic polysaccharides matrix. Lee et al. developed chitosan films incorporated with clove essential oil and HNTs. It was confirmed that the essential oils would be stabilized by HNTs without any surfactant. The addition of HNTs into the chitosan matrix enhanced the mechanical and water barrier properties of the chitosan films, and the active molecule, clove essential oil, imparted antimicrobial and antioxidant effects to the chitosan-HNTs nanocomposite films. The results showed that the films inhibited growth of mold which was derived from the surrounding environment. Therefore, the nanocomposite films could be used as active food packaging systems because of the antioxidant, antimicrobial properties and enhanced barrier property against water vapor [106]. (Lines452-488, Page 17-19, in the revised manuscript)

10) Line 404: add a comment on food packaging applications.

Reply: Thanks for the reviewer’s comment. We have added a comment on food packaging application in the revised manuscript. (Lines512-513, Page 19, in the revised manuscript)

Reviewer 2 Report

The article has the nature of the review and the authors must clearly highlight in the text. In addition my opinion of this manuscript is acceptable for publication after minor revision.

Author Response

Response to Reviewer 2 Comments

We are grateful for your comments and criticism. Indeed, your comments are very helpful for refining the original manuscript to a revised one. We took the following actions to your comments. We hope if these changes would be sufficient for publication of this paper in Polymers.

The article has the nature of the review and the authors must clearly highlight in the text. In addition my opinion of this manuscript is acceptable for publication after minor revision.

Reply: Many thanks for the reviewer’s suggestion. We review the recent progress toward the development of polysaccharide-HNTs composites, paying attention to the main existence forms, wastewater treatment and food packaging applications particularly. Through this review, we can have a better understanding of unique characteristics of polysaccharide-HNTs composites, which can be helpful to the continuous expansion of their application fields in the future. (Lines80-85, Page 3, in the revised manuscript)

Reviewer 3 Report

The paper provides an overview on recent advancements in the fabrication of nanocomposites based on halloysite nanotubes and polysaccharides. The topic falls within the scope of the journal. The review is well organized and written. I recommend its publication after the following revisions:

-  In the Introduction, the authors should indicate that the sizes of halloyiste depends on its specific geological deposit as reported in literature on the basis of microscopy [Applied Clay Science 74, 2013, 47-57] and scattering techniques [Applied Clay Science 160, 2018, 71-80]. Accordingly, it was demonstrated that the specific origin of halloysite nanotubes affects its reinforcing effect on polymeric matrix, such as pectin [ACS Appl. Mater. Interfaces, 9,2017, 17476–17488].

-  The scale bar in SEM images is not very clear and readable. Please check and revise.

-  In the paragraph 2.22.2 (Films), I suggest to report some examples regarding the preparation of functional nanocomposite materials obtained by the combination of polysaccharides and halloysite previously loaded with active molecules, such as antioxidants, antimicrobials and antiacids. Within this, I recommend to highlight that the vacuum pumping in/out procedure can optimize the halloysite loading from aqueous solution because of the water confinement mechanism. Recent literature on this mechanism should be quoted.

Bibliography. The format of some references (Refs. 11,56) is not correct. Please check and revise

Author Response

Response to Reviewer 3 Comments

We are grateful for your comments and criticism. Indeed, your comments are very helpful for refining the original manuscript to a revised one. We took the following actions to your comments. We hope if these changes would be sufficient for publication of this paper in Polymers.

1.  In the Introduction, the authors should indicate that the sizes of halloysite depends on its specific geological deposit as reported in literature on the basis of microscopy [Applied Clay Science 74, 2013, 47-57] and scattering techniques [Applied Clay Science 160, 2018, 71-80]. Accordingly, it was demonstrated that the specific origin of halloysite nanotubes affects its reinforcing effect on polymeric matrix, such as pectin [ACS Appl. Mater. Interfaces, 9,2017, 17476–17488].

Reply: Thanks for the reviewer for providing us these valuable references and suggestion. We have read these references carefully and found that they are closely related to our research. We have cited related works in the introduction part:[Applied Clay Science 74, 2013, 47-57](Lines553-554, Page 20, in the revised manuscript), [Applied Clay Science 160, 2018, 71-80] (Lines555-557, Page 20, in the revised manuscript) and [ACS Appl. Mater. Interfaces, 9,2017, 17476–17488] (Lines665-667, Page 20, in the revised manuscript).

2. The scale bar in SEM images is not very clear and readable. Please check and revise.

Reply: Thanks for the reviewer’s comment. We have checked and changed a high-definition image. (Lines267-268, Page 12, in the revised manuscript)

3. In the paragraph 2.2.2 (Films), I suggest to report some examples regarding the preparation of functional nanocomposite materials obtained by the combination of polysaccharides and halloysite previously loaded with active molecules, such as antioxidants, antimicrobials and antiacids. Within this, I recommend to highlight that the vacuum pumping in/out procedure can optimize the halloysite loading from aqueous solution because of the water confinement mechanism. Recent literature on this mechanism should be quoted.

Reply: Thanks for the reviewer’s suggestion. We have added the examples of nanocomposite film loaded with active molecules (Refs. 61,62) and quoted the latest literature on the water confinement mechanism (Refs. 63). A functional bionanocomposite film both with antioxidant and antimicrobial active molecules was successfully prepared by the filling of a pectin matrix with modified HNTs containing peppermint oil. Noteworthy, the prepared functional film was considered as a biocompatible material for packaging applications because of it was composed of eco-compatible molecules [61]. Makaremi, et al. developed functional films with antimicrobial properties that can be extended over time by dispersing HNTs/salicylic acid hybrid into the pectin matrix [62]. Moreover, it was demonstrated that the vacuum pumping in/out procedure can optimize the halloysite loading from aqueous solution because of the water confinement mechanism [63]. Accordingly, the attained knowledge can offer new routes in the preparation of effective delivery systems based on HNTs. (Lines224-234, Page 10, in the revised manuscript)

4. Bibliography. The format of some references (Refs. 11,56) is not correct. Please check and revise.

Reply: Thanks for the reviewer’s comments. We have checked and corrected the format of the reference 11 and 56.

Round  2

Reviewer 1 Report

The manuscript was changed accordingly to the suggestions and can be accepted for publication. Still some moderate English revision is needed. For instance,

“Mechanical and thermal properties of HNTs nanocomposite fibers were greatly enhanced, due to the uniform orientation of the well dispersed HNTs and greatly interactions  from cellulose and HNTs caused by hydrogen bonding, van der Waals and electrostatic interactions.”

Should be rewritten. since

“and greatly interactions  from cellulose and HNTs caused by hydrogen bonding, van der Waals and electrostatic interactions.” is non sense.

Reviewer 3 Report

The revised paper was improved according to the reviewers' suggestions. I recommend its publication in the present form.